# The Size of a Thyroid Nodule with ACR TI-RADS Does Not Provide a Better Prediction of the Nature of the Nodule: A Single-Center Retrospective Real-Life Observational Study

**DOI:** 10.3390/healthcare11121673

**Published:** 2023-06-07

**Authors:** Chiara Scorziello, Cosimo Durante, Marco Biffoni, Maria Carola Borcea, Fabrizio Consorti, Angelo Laca, Rossella Melcarne, Luca Ventrone, Laura Giacomelli

**Affiliations:** 1Department of Surgical Sciences, Sapienza University of Rome, Viale del Policlinico, 155, I-00161 Rome, Italy; marco.biffoni@uniroma1.it (M.B.); mariacarola.borcea@uniroma1.it (M.C.B.); fabrizio.consorti@uniroma1.it (F.C.); angelo.laca@uniroma1.it (A.L.); rossella.melcarne@uniroma1.it (R.M.); luca.ventrone@uniroma1.it (L.V.); laura.giacomelli@uniroma1.it (L.G.); 2Department of Translational and Precision Medicine, Sapienza University of Rome, Viale del Policlinico, 155, I-00161 Rome, Italy; cosimo.durante@uniroma1.it

**Keywords:** size, thyroid nodule, thyroid cancer, American College of Radiology, TI-RADS, ultrasound, predictive factor, real-world observational study

## Abstract

We investigated if thyroid nodule size has a predictive value of malignancy on a par with composition, echogenicity, shape, margin, and echogenic foci, and what would be the consequence of observing the rule of the American College of Radiology (ACR) to perform a fine-needle aspiration biopsy (FNAB). We conducted a retrospective real-life observational study on 86 patients who underwent surgery after a standardized diagnostic protocol. We divided the TR3, TR4, and TR5 classes into sub-classes according to the size threshold indicating FNAB (a: up to the threshold for no FNAB; b: over the threshold for FNAB suggested). We computed sensitivity, specificity, and positive (PPV) and negative predictive value (NPV) for the different sub-classes and Youden’s index (Y) for the different possible cutoffs. Each sub-class showed the following PPV (0.67, 0.68, 0.70, 0.78, 0.72), NPV (0.56, 0.54, 0.51, 0.52, 0.59), and Y (0.20, 0.20, 0.22, 0.31, 0.30). In this real-life series, we did not find a significant difference in prediction of malignancy between the sub-categories according to the size threshold. All nodules have a pre-evaluation likelihood of being malignant, and the impact and utility of size thresholds may be less clear than suggested by the ACR TIRADS guidelines in patients undergoing standardized thyroid work up.

## 1. Introduction

In the last decade, the detection of thyroid nodules on imaging studies increased, as did the number of thyroid cancer diagnoses, especially low-risk differentiated thyroid cancer [1]. This escalation of diagnoses has not been followed by a stable and corresponding increase in mortality rate for thyroid cancer. In fact, these cases are more often small thyroid cancers that are promptly treated with a good prognosis [2]. The dissociation between incidence and mortality has been attributed to the overdiagnosis and consequent overtreatment of non-aggressive small cancers [3,4]. Nevertheless, thyroid cancer is the most frequent (90% of cases) endocrine neoplasm and accounts for approximately 3.8% of all neoplasms. Thyroid cancer in Italy is the fourth most common malignancy in men up to the age of 50, the third most common malignancy in women up to the age of 50, and the fifth most common one in women between the ages of 50 and 70 [5].

These figures are a challenge for endocrinologists and surgeons, that should avoid the burden of unnecessary diagnostic work up or even surgery for diagnostic purposes, especially in low-risk patients, and, on the other hand, discover the nodules harboring a cancer to be treated in an early stage. Ultrasound (US) examination is the first-level exam to characterize thyroid nodules. Non-suspicious nodules are simply followed up over time and do not require further investigation. Otherwise, further diagnostic work up is necessary. Fine-needle aspiration biopsy (FNAB) helps the understanding of the nature of the lesion. However, approximately half of the nodules undergoing cytological examination are benign lesions and up to a third have indeterminate cytology [6]. International diagnostic guidelines now provide classification systems of the risk of malignancy of a thyroid nodule and guide the clinician in the most appropriate management of patients, selecting only the suspicious cases for further investigation and surgical treatment [7]. The American College of Radiology (ACR) TI-RADS system is meant to stratify the risk of malignancy of thyroid nodules, and its ability to discriminate between benign and malignant nodules is substantially greater than those of its competitors [8]. To reduce the number of unnecessary cytological examinations, TI-RADS sets a size threshold of the nodule to recommend a FNAB of nodules: for the mildly suspicious nodules the threshold is rather high (>2.5 cm), 1.5 cm for moderately suspicious nodules, and 1 cm for highly suspicious ones [8]. Hence, the size of the nodule is considered only after the assessment of the likelihood of malignancy, as an indicator to perform FNAB on suspicious nodules, but does not have a role in the calculation of the predictive value, even if—implicitly—smaller nodules are expected less likely to be a tumor.

The aim of this real-life observational study is to answer the following research questions:
Does the size of the nodule have a predictive value on a par with composition, echogenicity, shape, margin, and echogenic foci?What are the consequences of using a high size threshold to suggest FNAB? The hypothesis is that using the high size threshold of a nodule to suggest FNAB, as indicated by the ACR TIRADS guideline, will miss a significative number of tumors.

## 2. Materials and Methods

### 2.1. Study Design

We reviewed our surgical pathology files over the period from 1 November 2015 to 30 April 2020. We identified 482 cases that underwent thyroidectomy or hemithyroidectomy with or without central or lateral cervical lymphadenectomy. Our Thyroid Unit adopts a standardized diagnostic procedure: every patient referred for a nodular disease of the thyroid meets an endocrinologist who can make an indication for surgery after considering the size of the nodule, the status of the cervical nodes, any symptoms reported by the patient (sense of “encumbrance” in the anterior neck region, dysphagia, dyspnea), the US examination, and the cytological examination of the suspicious nodule. The patient is then referred to the endocrine surgeon to confirm the indication for surgical treatment. Every decision is made after an open discussion with the patient about all the possible treatment options—including follow up—and their informed consent. All the operations were made by the same team of two accredited surgeons of the Thyroid Unit.

Patients with a single nodule ranging in size from 1 to 4 cm, without lymph node involvement, were referred for either hemithyroidectomy or total thyroidectomy surgery, also considering their preference. Patients with nodules larger than 4 cm or extracapsular extension, multifocal lesions, and clinically positive nodes underwent total thyroidectomy and, if necessary, lymph node dissection of the central compartment and/or unilateral or bilateral cervical dissection.

Usually, the endocrinologists perform the US, classifying the nodules according to the ACR TIRADS system. The final assessment is formulated only after the agreement of two clinical examiners to decrease inter-operator variability which characterizes the US examination [9]. To obtain a valid estimate of the power of prediction of malignancy, we included in the study only patients who had undergone the whole procedure, including the US investigation according to the procedure described above, with a 13 MHz linear probe with “HI VISION AviusR US System”. The characteristics of the nodules, such as the diameter (the nodule was defined as “taller than wide” whenever the ratio between the anteroposterior diameter and the transverse one was ≥1), the margins, the structure and composition, echogenicity, the presence of calcifications or other hyperechoic foci, and any extrathyroidal extensions, were evaluated. The predictive US score of malignancy was calculated for each examined nodule according to the risk stratification systems of the ACR TI-RADS. FNAB was indicated for all solid nodules, including TIRADS2. FNAB was performed with a fine needle (23–25 gauges) and under US guidance. The result of the cytological examination was classified in accordance with the criteria established by the Italian Classification of Thyroid Cytology [10]. After the surgical removal of the gland or a part of it, the nodules underwent definitive histological examination at the pathology department. The same pathologist assessed all cases both for cytology and surgical specimen. Data were prospectively collected in an electronic database. 

The exclusion criteria were: (1) clinical data were incomplete; (2) nodules did not have a preoperative study according to the procedure described above and/or had no available cytological results; and (3) nodules did not undergo definite histological examination from the pathologist of the Thyroid Unit (lack of gold standard). Figure 1 shows the flow of patients’ selection.

### 2.2. ACR TIRADS

In 2017, the ACR proposed a new version of TI-RADS [11]. According to the ACR classification, the composition of the nodule (cystic, spongiform, solid), the echogenicity (anechoic, hyperechoic, hypoechoic, isoechoic), the shape (wider than tall, taller than wide), the margins (smooth, regular, irregular, extra thyroidal extension), and the presence of echogenic foci (none or large comet-tail artifacts, macrocalcifications, peripheral calcifications, punctate echogenic foci) are crucial for the evaluation of the prediction of malignancy; the system assigns a score from 0 to 3 for each of these characteristics. The sum of these scores constitutes the total score indicative of the increasing risk of malignancy according to five classes [12] (Table 1).

Finally, since there is not a reporting guideline for observational real-life studies, we used the RELEVANT checklist for the assessment of quality [13].

### 2.3. Statistical Methods

The power of the study was calculated with the Open Epi calculator ver. 3.01, considering a prevalence of thyroid nodules in 30% of the general population and a prevalence of cancer in 5% of the nodules [14]. For a confidence level of 95%, 73 cases were needed. A recent, wide, long-term cohort study [15] reported a much lower prevalence of 1.2%. If we assumed this expected frequency, the number of needed cases would be even lower.

We split the TR3, TR4, and TR5 classes of the ACR TI-RADS system according to the size threshold indicating FNAB (a: up to the threshold for no FNAB; b: over the threshold for FNAB suggested—see Table 1). Assuming the final pathological report as a golden standard for diagnosis, we computed sensitivity (SE), specificity (SP), and positive (PPV) and negative predictive value (NPV) for the different resulting sub-classes. To compute the effect of including the size as a predictive information, we also iteratively calculated the Youden’s index (Y statistics) [16] for different possible cutoffs. 

The differences in the rate of true/false positive and true/false negative for the different thresholds were evaluated with the Fisher exact test, with an alpha error <0.05.

### 2.4. Ethical Statement

This is a retrospective real-life observational study. According to Italian regulation, studies are allowed to use anonymized and aggregated data without further explicit consent from the patients, because the consent to scientific statistical use of routinary clinical data is included in the informed consent at the beginning of the clinical process.

## 3. Results

Eighty-six patients (62 females; 24 males) were finally enrolled in the study (Figure 1) with mean age of 49.8 ± 14.5 (SD) years. Seventy-nine patients underwent total thyroidectomy and seven lobectomies, five central lymphadenectomies, nine latero-lateral lymphadenectomies, of which three were bilateral, and twenty-four biopsies of lymph nodes removed as suspicious during intraoperative control. Eighty-nine thyroid nodules were analyzed. Malignant lesions were 47 papillary thyroid cancer (PTC), 3 medullary thyroid cancer (MTC), 3 follicular thyroid cancer (FTC), and 1 poorly differentiated carcinoma (PDTC). Table 2 shows more details on the final pathology.

The benign lesions were 23 multinodular hyperplasia nodules, 1 hyperplastic thyroid nodule in the context of chronic lymphocytic thyroiditis, 8 microfollicular adenomas, 2 adenomatoid nodules, and 1 colloid-hemorrhagic nodule. The ACR TI-RADS score classified the 89 nodules as TR2 (15, 16.8%), TR3 (15, 16.8%), TR4 (22, 24.8%), and TR5 (37, 41.6%). Table 3 shows the distribution of benign and malignant nodules for each sub-class of the ACR TI-RADS, according to the size of the nodule.

Table 4 reports the diagnostic performance of an arbitrary threshold (TR2 and TR3: non-suspicious nodules; TR4 and TR5: suspicious nodules) without any further subdivision according to the size of the nodule. The table also reports the performance of different thresholds, according to the “a” and “b” sub-classes for TR3, TR4, and TR5 classes. Table 5 shows results of cytology from US-guided fine-needle aspiration for each TIRADS category.

## 4. Discussion

Our data, obtained from a real-life series, do not support a predictive value of the size of the nodule. In fact, there was no significant difference in PPV and NPV between the “a” and “b” sub-classes of the ACR TIRADS classes. Table 4 shows that the confidence intervals of SE, SP, PPV, and NPV largely overlap among the different thresholds. This result indicates that the size does not have a significant predictive value, but also that, although the overall dimension of the sample gave the study enough power, splitting the whole series of 89 nodules into the sub-categories, the number of nodules in each category was too low to warrant a narrower interval of confidence. More importantly, as to the second research question, using the thresholds suggested by the ACR TIRADS guideline would have led us to miss a significant number of neoplasms. If we had not adopted a more extensive use of cytology than indicated by the ACR TIRADS guideline, 20 out of 71 TR3-TR5 (28.2%) cancer diagnosis would have been missed (Table 3): 7/10 TR3a nodules, 5/9 TR4a nodules, and 8/8 TR5a nodules.

The association between nodule size and malignancy is controversial. Some authors suggest that nodule size is inversely related to thyroid cancer, and nodules with a higher diameter are less likely to be malignant [17]. However, the prognosis of some tumors (such as follicular or Hurthle cell carcinoma) is related to the size of the nodule; a thyroid carcinoma smaller than 5 mm compared with a 6–10 mm diameter has a better survival rate and less recurrence at 5 years [18]. In US scores, such as EU TIRADS or ACR TIRADS, size is used to discriminate which nodule, with a specific US score, must be biopsied. Nevertheless, size is considered only after the attribution of a risk grade based on other nodules’ US features. In our analysis, we considered size as a US feature, on a par with composition, echogenicity, shape, margin, and echogenic foci, to be added to ACR-TIRADS score, and we could not observe a significant difference between standard categories and the sub-categories corrected by nodule size (Table 4). The final pathology confirmed that the rates of malignancy were the same regardless of nodule size, hence the quantitative parameter of the size of the nodule in this series did not increase the predictive value of preoperative US examination. The highest value of Y statistics was between TR4b and TR5a, as well as TR5a and TR5b, which is not surprising since the TR5 class yields the highest probability of malignancy. Nevertheless, none of the differences in true/false positive and negative were significant (Table 3).

In partial agreement with our findings, in a comparison of a back-propagation neural network (BPNN) with a multivariate logistic regression model to predict malignancy in thyroid nodules [19], size was inversely correlated in both models with the likelihood of malignancy. On the contrary, in their study on 1044 nodules, Strieder et al. [20] systematically performed FNAB and found that, in applying the ACR TIRADS guideline, they would have missed only 0.9% of malignant cases. Nevertheless, their conclusion is based on the cytology report and not on post-surgery pathology, and they considered as “malignant” only Bethesda five and six classes, so it is likely that the number of missed malignancies was higher. In the ACR white paper [11], the motivation to limit the number of “unnecessary” FNAB is that “diagnosing every thyroid malignancy should be not our goal” and that there is an “increasing trend toward active surveillance”. Small neoplastic nodules not subjected to FNAB are left to grow until reaching the threshold, and this could delay the diagnoses by months or years, although this would not have an impact on survival. In a cohort study in Japan [21], 1235 patients with low-risk papillary microcarcinomas chose observation without immediate surgery. The range of observation was from 18 to 227 months: 191 were operated on, and none of the 1235 patients showed distant metastasis or died of PTC during observation. This is the kind of argument supporting the claim of overdiagnosis, leading to unnecessary treatments. Nevertheless, the success of this strategy also depends on the patient’s compliance to follow up, which cannot be taken for granted in a real-life situation [22]. In fact, our real-life study also calls attention to patients’ reactions and behavior. 

Every human behavior is also culturally determined. In a qualitative study on Australian patients’ experiences of diagnosis and management of papillary thyroid microcarcinoma [23], patients’ preference for treatment was largely based on eliminating the possibility of the cancer spreading (thyroidectomy) or not wanting to be on thyroid replacement medication (preference for hemi-thyroidectomy). In a mixed-method study on the risk attitude of Italian patients [24], most of the participants showed to be risk averse. We are not questioning the increasing evidence that many differentiated thyroid cancers are not a serious threat and can be monitored over time, without an increase in mortality [3,4]. This evidence gives a strong foundation of knowledge to the process of clinical decision. We are posing the issue of the process of shared decision of care, in which the provision of information and the way in which information is provided is crucial. [25] The “framing effect” refers to the different outcomes a communication has if framed in a positive or negative framework.

There is not enough evidence supporting the size threshold for carrying out an FNAB [26]. In addition, a small nodule does not necessarily mean a less invasive one. For example, there is growing evidence that the mutational status may supersede the size as a prognostic factor. [27] Furthermore, some studies report that cervical lymph node metastases are positively related to the size of the primary tumor and, the larger the tumor is, the higher the incidence of cervical lymph node metastases. Although there is not a universally accepted threshold size value, many authors agree that a tumor size larger than 1.0 cm is an independent risk factor for cervical lymph node metastasis, ranging from 20% to 66% [28]. This is associated with a higher risk of recurrence and affects the patient’s quality of long-term life. In addition, together with the incidence of metastasis, even tumor dedifferentiation would increase along with tumor survival and aging [29,30].

All nodules of the thyroid have a pre-evaluation likelihood of being malignant and, for those ones with the appropriate constellation of clinical and radiographic findings, fine-needle aspiration biopsy should be proposed, regardless of size [31,32]. Finally, although many studies advocate that ACR TIRADS classification has the best diagnostic performance, especially in terms of lowest percentage of unnecessary biopsies [8,33], an Italian consensus defined lower thresholds to recommend FNAB, according to different US features [14].

In our series, we had a higher rate of malignant nodules in TR3, TR4, or TR5 categories than in the literature [34], but this is a consequence of the selection effect with positive cytology. This finding also highlights the importance of considering the final pathology after surgery as the most relevant outcome. 

In the era of precision medicine, it is important and useful to find a way to correctly identify malignant thyroid nodules, reducing the diagnostic pressure of overtreatment that can expose patients to unnecessary medical procedures, frequently invasive ones [35]. International guidelines advocated that treatment strategies for patients must be tailored to individual patient’s needs to guarantee that the benefits outweigh the risks of adverse outcomes [36,37]. We endorse the idea that the availability of more information to support a shared decision-making process could be a gain commensurate with the cost of performing more FNABs.

The study has some limitations. First, despite the fact that overall number of the series gave the study enough power, when we split the cases into the “a” and “b” sub-groups according to the TIRADS and size, some sub-groups were very small. A selection bias may also be present because we retrospectively only included thyroid nodules which underwent surgery after completing the described diagnostic protocol, and this has probably led to a higher number of malignant nodules. Another limitation of the study, which can be defined as intrinsic to the US method itself, is represented by the discriminative capacity of the US evaluation of the nodule. A strength of the study is the possibility of comparing the initial risk stratification based on US examination, summarized in the ACR TIRADS score, with the result of the post-surgery pathology, as the gold standard of the study.

Moreover, this article shows the weakness, but also the strength, of a real-life study. To rely on accountable data, we had to reduce the overall series of 486 clinical cases to only 86 (17.7%). Nevertheless, in this selected sub-group of patients we could show that some patients would have missed the opportunity to rely on more information about the nature of their nodule to make a decision.

## 5. Conclusions

The result of this study adds an important element to the current knowledge, showing how different the context of care can be in a real setting, when the correctness of clinical performance is granted by a standardized protocol, compliant with international guidelines and integrated with a more global clinical evaluation, that also encompasses patient’s preference and their propensity to undergo a strict follow up or surgery. 

The impact and utility of size thresholds may be less clear than suggested by the ACR TIRADS guidelines in patients undergoing standardized thyroid work up. This could drive medical and patient education and the design of techniques and new shared decision-making tools to improve conversations among clinicians, in a multidisciplinary perspective, and between clinicians and patients on treatment options. 

## Figures and Tables

**Figure 1 healthcare-11-01673-f001:**
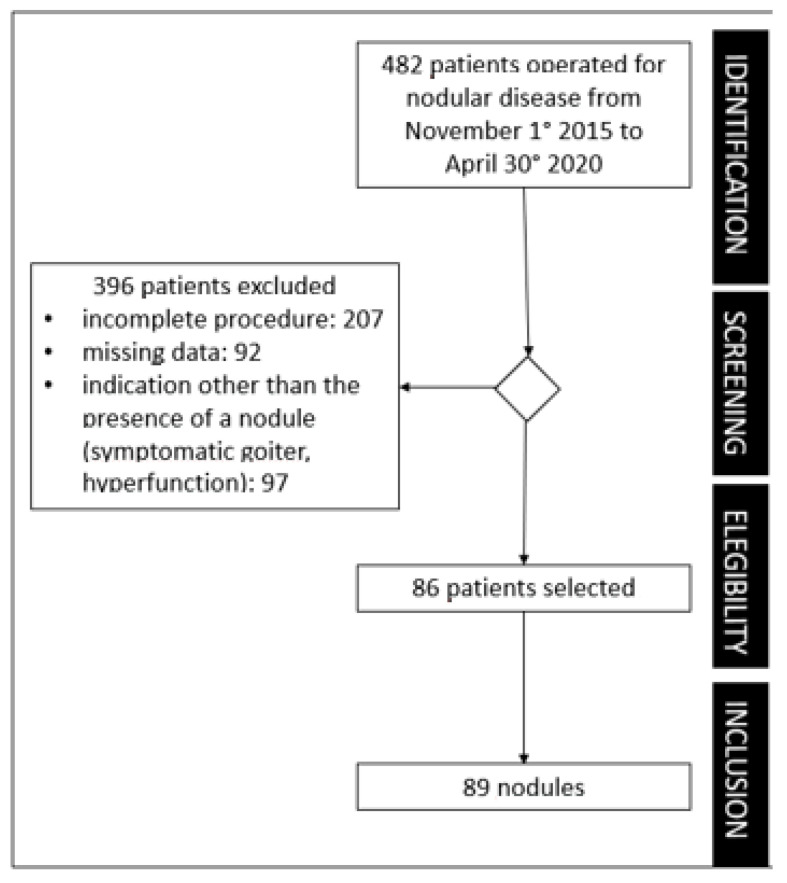
Flowchart of the study.

**Table 1 healthcare-11-01673-t001:** The ACR TI-RADS recommends FNAB for highly suspicious nodules 1 cm or larger; the thresholds for mildly suspicious and moderately suspicious nodules are 2.5 and 1.5 cm, respectively. (Source: ACR White Paper 2017 [12]).

Summation of Points from Each Column to Determine Ti-Rads Grade				
0 Points	2 Points	3 Points	4 to 6 Points	7 to More Points
TR1 Benign	TR2 Not Suspicious	TR3 Mildly Suspicious	TR4 Moderately Suspicious	TR5 Highly Suspicious
No FNA	No FNA	≥2.5 cm FNAB	≥1.5 cm FNAB	≥1.0 cm FNAB

**Table 2 healthcare-11-01673-t002:** Distribution of histological types: T and N staging in 86 patients.

Variables	No. (%)
Histology	
- benign - papillary classical variant - papillary follicular variant - papillary incapsulated - follicular - medullary - Poorly differentiated	35 (39.3) 40 (44.9) 6 (6.7) 1 (1.1) 3 (3.4) 3 (3.4) 1 (1.1)
Tumor size	
- T1a - T1b - T2 - T3 - T4	30 (55.6) 15 (27.8) 5 (9.2) 3 (5.6) 1 (1.8)
Nodal metastasis	
- N0 - N+	37 (68.5) 17 (31.5)

**Table 3 healthcare-11-01673-t003:** Classification of the nodules according to the ACR TIRADS system. All the differences between the “a” and “b” sub-classes were not significant (Fisher exact test).

ACR TIRADS	Number of Nodules	Number of Benign Nodules	% of Benign Nodules	Number of Malignant Nodules %	% of Malignant Nodules
TR2	15	11	73.33	4	26.67
TR3a	10	3	30.00	7	70.00
TR3b	5	2	40.00	3	60.00
TR4a	9	4	44.44	5	55.56
TR4b	13	7	53.85	6	46.15
TR5a	8	0	0	8	100.00
TR5b	29	8	27.59	21	72.41

**Table 4 healthcare-11-01673-t004:** First row: diagnostic performance [sensitivity (SE), specificity (SP), positive (PPV) and negative predictive value (NPV), Youden’s index (Y), confidence interval at 95% (CI 95%)] of ACR TIRADS with a threshold set between TR3 and TR4 (mildly and moderately suspicious nodules) intended as negative and positive test. Following rows: diagnostic performance for different thresholds.

**Cut-off: TR3–TR4**								
**SE**	**CI 95%**	**SP**	**CI 95%**	**PPV**	**CI 95%**	**NPV**	**CI 95%**	**Y**
0.74	0.62–0.86	0.46	0.29–0.62	0.68	0.56–0.80	0.54	0.35–0.71	0.20
**Cut-off: TR3a–TR3b**								
**SE**	**CI 95%**	**SP**	**CI 95%**	**PPV**	**CI 95%**	**NPV**	**CI 95%**	**Y**
0.80	0.69–0.90	0.40	0.24–0.56	0.67	0.56–0.79	0.56	0.36–0.75	0.20
**Cut off: TR4a–TR4b**								
**SE**	**CI 95%**	**SP**	**CI 95%**	**PPV**	**CI 95%**	**NPV**	**CI 95%**	**Y**
0.65	0.52–0.77	0.57	0.41–0.73	0.70	0.57–0.83	0.51	0.36–0.67	0.22
**Cut off: TR4b–TR5a**								
**SE**	**CI 95%**	**SP**	**CI 95%**	**PPV**	**CI 95%**	**NPV**	**CI 95%**	**Y**
0.54	0.40–0.67	0.77	0.63–0.91	0.78	0.65–0.92	0.52	0.38–0.65	0.31
**Cut off: TR5a–TR5a**								
**SE**	**CI 95%**	**SP**	**CI 95%**	**PPV**	**CI 95%**	**NPV**	**CI 95%**	**Y**
0.52	0.37–0.70	0.77	0.63–0.91	0.72	0.56–0.89	0.59	0.44–0.73	0.30

**Table 5 healthcare-11-01673-t005:** Results of cytology from US-guided fine-needle aspiration for each TIRADS category.

ACR TIRADS	TIR1	TIR2	TIR3A	TIR3B	TIR4	TIR5
TR2	1	7	1	4	2	0
TR3	0	3	4	3	4	0
TR4	0	9	0	9	3	1
TR5	1	3	4	6	11	12

## Data Availability

The data presented in this study are available on request from the corresponding author.

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
