# Peer review of "The Size of a Thyroid Nodule with ACR TI-RADS Does Not Provide a Better Prediction of the Nature of the Nodule: A Single-Center Retrospective Real-Life Observational Study"

_healthcare, 2023, doi:10.3390/healthcare11121673_

Round 1

Reviewer 1 Report

Dear Authors,

The paper is interesting and I believe it adds value to our current knowledge. The study of thyroid nodules is extremely important due to the major epidemiologic impact of such lesion. The real-life medicine studies are mandatory since they reflect different aspects that are not otherwise captured in controlled studies.

I only have a few minor observations, suggestions, recommendations or questions, as following.

1.    Title. No need for point at the end of the title.

2.    Title. I suggest to re-organize the title since you start with a question and then you introduce the study design (that seems like the answer to your question).

3.    Abstract. You start with a verb – I believe you intended to mention first “aim” or “purpose” or “objective” or “end point”.

4.    Abstract. Please re-organize the presentation of the abstract. First the inclusion/exclusion criteria according to the study design followed by major Results followed by a Conclusion which should be strictly related to these Results.

5.    Introduction. I suggest to start with a general statement concerning first the thyroid nodules and then the risk of underlying a thyroid cancer since your study starts from thyroid nodules. Also, the specific data from Italy are useful, but, in order to capture the readers’ interest I suggest starting with some general data followed by specific data concerning Italy.

6.    Line 47. Please use “help the understanding”.

7.    Line 56. No “US”, please, use instead “to stratify the risk”.

8.    Methods. The first section is simply the study design.

9.    Methods. The second section involves the inclusion/exclusion criteria of enrolled patients. The final number of the patients you achieved only after applying this methodology thus it is rather a part of Results.

10. Are there any ethical concerns to be mentioned at Methods?

11. At the end of each Table, please re-introduce the abbreviations according to each table.

12. Results – What do you mean by “aesthetic reason” or by “symptomatic nodules”?

13. Who performed the thyroid surgery? Was it the same surgeon or team of surgeons?

14. Who analyzed the post-operatory pathologic reports and FNAB – based cytological reports? Was it the same team?

15. Conclusion. This is a very simple take-home message based on your original data, and maybe your study - based conclusion to further expansion of the research in order to specifically address the current knowledge. No references are allowed at this section.

Thank you

None

Reviewer 2 Report

I appreciate your enthusiasm for thyroid nodules. But I have some comments. 

  1. English must be improved even for better understanding as for grammatical correctness.

  2. I miss the point of the study, because the size limit indicating FNA has nothing to do with prediction of malignancy. All these US systems are made to not miss clinically important TC and in contrast not to examine all TC. As we know from autopsies especially microTC are very common not harming the patients and the prognosis of these “indolent” carcinomas is excellent (99.5%). The thyroid nodule sizes indicating FNA have some reasons? You should check it out. For example, the active screening of TC was called off in South Korea, because of no benefits. Also I miss the thyroid nodule sizes in your study. 

  3. line 16 ACR TI-RADS diagnostic hypothesis? What is it?

  4. line 66-87 (excluding 75-79 line) belongs to methods, also table 1

  5. line 86 RELEVANT assessment? What is it?

  6. line 104 aggressive histological variants, but you usually know the results after surgery?

  7. line 108 standardized procedure? What is it?

  8. line 114 taller than wide, there is a definition of AP to transverse dimension ≥1

  9. Figure 1 must be improved, for example inclusion and exclusion criteria, 89 thyroid nodules in 86 patients underwent histological confirmation….

  10. You don't need to explain Younden´s index and it is better to write Younden and not J statistics. 

  11. Table 2 you don't need to repeat the description of TR groups, already in methods, but you should describe the “a” and “b” groups.

  12. pick up the same description of the groups TR3A or TR3a

  13. sometimes you use in numbers “comma” and sometimes “dot”. It must be the same. 

  14. Table 2 is the same as Table 3, but in other words.

  15.  Table 3 and Table 4 maybe make just one Table, correct is CI 95%, check which abbreviations are usually used for sensitivity, Younden´s etc. 

  16.  Figure 2 it is not necessary.

  17. line 224 negatively affecting survival curve..It is just an author opinion, not proven by study; citation 19.

  18. line 269-277 it is not necessary to discuss. 

  19. Limits: sample size, too small. 

  1. English must be improved even for better understanding as for grammatical correctness.

Reviewer 3 Report

In general, the research question to adress the impact of nodule size when applying ACR TIRADS is of interest and clinical importance.  

The abstract refers to the different thresholds calculated but restricts to results for TR3/TR4, only. This sentence alone provides no usefuel information to the reader. 

The abstract and discussion section suggests that  nodule size thresholds do not provide incremental predictive value when applying ACR TIRADS for FNA selection - a statement that can not be concluded from the limited amount of data. Please refer to the methodological criticism given below.

Neither the introduction section nor, later on, the conclusion section do mention or discuss the well accepted phenomenon of thyroid cancer overdiagnosis (Lancet 2022 DOI:https://doi.org/10.1016/S2213-8587(22)00035-3), NEJM 2016 https://www.nejm.org/doi/full/10.1056/NEJMp1604412). Aggressive nodule screening and work-up increase the prevalence and incidence of thyroid cancer (TC) without affecting overall TC mortality - this should be commented on. 

Methods: The sample of 86 patients selected out of 482 TC patients represents just a small proportion and might be a source of bias. Please comment on screening and study collective with respect to baseline T stage, histology, and reasons for referral to surgery. 

Sample size calculation: The assumed cancer prevalence in thyroid nodules was assumed to be 5%, which seems very high if any nodule detectable by US had been considered, since the authors state a 30% prevalence of thyroid nodules in the population. In fact, the Italian Consensus refers to, but does not provide orignial data (12).  Please comment on.   

Results: In total the findings are based on 54 TC patients. PTC accounted for more than 85% of TC cases in the study population suggesting some selection bias. Please comment on the distribution pattern of histopathologic variants in these PTC, in particularly the exclusion of encapsulated forms.

The confidence intervals given in table 4 do overlap for subgroups - please comment on.

The Youden´s index is low in all subgroups but relates to a minimum of 3 TC cases in group T3B questioning the validity of the findings. 

The conclusion section is not focused on the research question - paragraphs on AI, patient-physician relationship or EBM should be removed.

References should be given for the statement on line 232  

please change or condense or modify:

sentences starting with line 24

line 37 modify `increase of diagnosis` and balanced and equivalent increase* in 

line 93 should better read for follow-up 

line 100 should read by the patient

line 225 modify future survival curve

line 226 modify sentence

Round 2

Reviewer 2 Report

I appreciate improvement of the article.

Just little comments:

line 67 Does have and not has the size of the nodule a predictive value on a par with composition, echo- 67 genicity, shape, margin, and echogenic foci?

Materials and Methods: usually starts with study design (2.1), 2.2 ACR TI-RADS instead of context? 

Figure 1 Flowchart of the study?Instead of patient´s selection. 

Line 144 splitted and not spitted.

List above in comments. 

Reviewer 3 Report

The paper has been significantly improved with regard to the introduction, methods and discussion section. However, the last sentence of both abstrat and conclusion section do not reflect what has been found and should be therefore modified. The former should better read that the impact and usefulness of size tresholds might be less clear then suggested by ACR TIRADS guidelines in patients undergoing standardized thyroid work up in an Italian tertiary center. Comments on overdiagnosis should be omitted in the abstract as on speculative, culturally related mechanisms in the last paragraph of conclusion section. 

improved but editing required
